# Vitamin C Improves Oocyte In Vitro Maturation and Potentially Changes Embryo Quality in Cattle

**DOI:** 10.3390/vetsci11080372

**Published:** 2024-08-13

**Authors:** Yueqi Wang, Aibing Wang, Hongmei Liu, Rui Yang, Boyang Zhang, Bo Tang, Ziyi Li, Xueming Zhang

**Affiliations:** 1State Key Laboratory for Diagnosis and Treatment of Severe Zoonotic Infectious Diseases, Key Laboratory for Zoonosis Research of the Ministry of Education, College of Veterinary Medicine, Jilin University, Changchun 130062, China; yueqiw22@mails.jlu.edu.cn (Y.W.); wangaibing1023@163.com (A.W.); lhm2017851002@163.com (H.L.); ruiyang22@mails.jlu.edu.cn (R.Y.); zby23@mails.jlu.edu.cn (B.Z.); tang_bo@jlu.edu.cn (B.T.); 2First Hospital, Jilin University, Changchun 130021, China

**Keywords:** cattle, embryo, in vitro maturation, oocyte, vitamin C (VC)

## Abstract

**Simple Summary:**

Assisted reproductive technology including in vitro maturation (IVM) of oocytes, in vitro fertilization (IVF), and in vitro culture (IVC) of embryos is widely used both for humans and animals. In animal science and veterinary medicine, it is particularly important for livestock breeding. Although numerous efforts have been made, the development rates obtained with these techniques are still limited, especially in cattle. Vitamin C is a key nutrient that has multiple functions. To procure high-quality bovine oocytes, the effects of vitamin C on the IVM of bovine oocytes and embryo development at an early stage were investigated. We found that vitamin C supplementation at a tuned concentration improved bovine oocyte maturation and its combination in the media of IVM and IVC can potentially change the quality of bovine embryos.

**Abstract:**

To obtain high-quality bovine oocytes, the effects of vitamin C (VC) on the IVM of bovine oocytes and early embryo development were investigated. The results showed the following. (1) The IVM medium containing 50 µg/mL VC improved the oocyte maturation rate but did not affect the parthenogenetic embryo development. (2) The IVC medium containing 20 µg/mL VC improved the cleavage rate of the IVF embryos and enhanced the mRNA transcriptions of pluripotency gene *Oct4*, *Sox2*, *Cdx2,* and *Nanog* in the blastocysts but had no effects on the blastocyst rate. (3) Combining supplementation of 50 µg/mL VC in IVM medium + 20 µg/mL VC in IVC medium (named as VC 50/20, similar hereinafter) elevated the cleavage rate of IVF embryos and enhanced the mRNA expressions of *Oct4*, *Sox2*, *Cdx2,* and *Nanog* in the blastocysts. (4) Combination of VC 0/20 and VC 50/20 enhanced the transcription of anti-apoptotic gene *Bcl-2* and VC 50/0 weakened the transcription of pro-apoptotic gene *Bax*, while VC 0/40 and VC 0/60 increased *Bax* expression and diminished the *Bcl-2*/*Bax* ratio in blastocysts. Together, employing 50 µg/mL VC improves the IVM of bovine oocytes and combination of VC 50/20 potentially changes bovine embryo quality by enhancing the expressions of the pluripotency genes and regulating the expressions of apoptosis-related genes.

## 1. Introduction

The in vitro maturation (IVM) of oocytes and subsequent early embryo development are two key steps in assisted reproductive technology, both for treatment of human infertility and for high-yield, top-quality, and disease-resistant livestock breeding. In order to increase the oocyte IVM rate and embryo development rate, great efforts have been made and multiple parameters were found to be involved in these processes [1,2]. However, the embryo quality and development rates obtained with these techniques are still limited, particularly in large domestic animals like cattle.

Vitamin C (VC) is also named Ascorbic Acid/L-Ascorbic Acid. It is a necessary nutrient that needs to be consumed in sufficient levels through food in order to prevent hypovitaminosis C, its repercussions, and the potentially deadly deficiency condition scurvy. As a water-soluble molecule, VC is well known for its multiple functions such as the promotion of wound healing, detoxification, alleviation of diabetes and neurogenic/synaptogenic disease, epigenetic regulation, serving as a cofactor for various enzymes, cytotoxicity against tumor/cancer cells, and antioxidant activity [3,4]. Among these functions, VC contributes electrons to a variety of enzymatic and non-enzymatic processes to carry out its roles in collagen production, vasculogenesis, aging, cell proliferation, and differentiation [5]. Recently, it has been demonstrated that treatment with VC improved the development of mouse IVF embryos [6] and the supplements of VC as well as other molecules seem to be also beneficial to the development of bovine embryos [7]. Later studies have shown that VC-treated bovine donor cells increased the developmental capacity of the cloned embryos by regulating embryonic transcription [8]. It has also been reported that VC improved the developmental capability of swine oocytes after parthenogenetic activation and somatic cell nuclear transplantation [9] and enhanced the survival of primary ovarian follicles cultured in vitro [10]. However, its appropriate concentration still needs to be determined for the IVM of bovine oocytes and embryo development. Additionally, the effects of the combined application of VC both in oocyte IVM and embryo in vitro culture (IVC) of cattle are unclear.

During the early development of embryos, the divergent pluripotent states of the embryonic cells and the stepwise pluripotency transitions of embryonic stem cells (ESCs) are closely related to the expressions of pluripotency genes [11,12]. Along with embryonic cell proliferation and differentiation, cell apoptosis might also occur [13,14]. The transcription level of pluripotency genes and apoptotic-related genes can also reflect the embryo’s quality. Therefore, to address the questions mentioned above, we examined the effects of VC and its combinations in IVM medium and IVC medium on the IVM of bovine oocyte and early embryonic development by employing methods of morphological assays and transcriptional evaluation of the common pluripotency-related genes (*Oct4*, *Sox2*, *Cdx2*, and *Nanog*) and apoptotic-related genes (*Bcl-2* and *Bax*), aiming to increase the efficiency of embryo IVC and procure high-quality embryos in cattle.

## 2. Materials and Methods

### 2.1. Bovine Ovaries and Chemicals

Ovaries were obtained from a nearby abattoir (Changchun Jixing meat industry Co., Ltd., Changchun, China) and delivered to the laboratory in a saline solution containing 200 IU/mL penicillin at 36–37.5 °C within 2 h. All experimental materials and procedures used here received endorsement from the Animal Welfare and Research Ethics Committee at Jilin University (number of permit: SY201903002). Unless stated otherwise, chemicals and reagents were all procured from Sigma-Aldrich. The basic medium was acquired from Gibco (Oakland, CA, USA).

### 2.2. IVM

The follicles (2–8 mm in diameter) were selected under a stereomicroscope. Oocytes were aspirated by using a 10 mL syringe fitted with a 12-gauge needle. The cumulus-oocyte complexes (COCs) containing intact and compact cumulus cells were selected. The COCs were rinsed three times with maturation solution [TCM-199 enriched with 25 mmoL/L NaHCO_3_, 0.38 mmoL/L sodium pyruvate, 10 μg follicle-stimulating hormone (FSH)/mL, 10 μg luteinizing hormone (LH)/mL, 1 μg 17β-estradiol (E2)/mL and 10% (*v*/*v*) fetal bovine serum (FBS)]. Next, the COCs were transferred into 100 µL maturing solution microdroplets (10–15 COCs/droplet) and the IVM was conducted at 38.5 °C, 5% CO_2_, and saturated humidity for 20–22 h. VC (L-Ascorbic Acid, A4403, Sigma-Aldrich, St. Louis, MO, USA) at concentrations of 0, 25, 50, and 100 µg/mL was added into the maturing solution based on our preliminary experiments. The maturation rate was calculated and statistically analyzed after the oocytes had been cultured for 23–24 h. After IVM, the oocyte with the first polar body, homogeneous cytoplasm, and at least three layers of cumulus cells was deemed mature and of high morphological quality.

### 2.3. Parthenogenetic Activation

The cumulus cell-free IVM oocytes were obtained by the following steps. Firstly, they were digested for 2–3 min in the operating medium (15 mmol/L Hepes, 5 mmol/L NaHCO_3_, and 3 mg/mL of bovine serum albumin) by supplementation with 0.2% (*w*/*v*) hyaluronidase. Post-digestion, the cumulus cells were repeatedly and gently blown away with a 200 µL pipette and the oocytes were rinsed 3 × 5 min with the operating medium. After the collection of the mature oocytes, parthenogenesis is activated by a sequential treatment of ionomycin and 6-dimethylaminopurine (6-DMAP). Briefly, the oocytes were first incubated with an operating solution containing 5 µmol/L ionomycin for 5 min. Next, the eggs were delivered into the modified synthetic oviductal fluid (mSOF) droplets enriched with 2 mmol/L 6-DMAP and cultured at 38.5 °C, 5% CO_2_, and saturated humidity for 4 h. The mSOF consists of 108 mM NaCl, 7.2 mM KCl, 0.3 mM KH_2_PO_4_, 5 mM NaHCO_3_, 3.3 mM sodium lactate, 0.07 mM kanamycin monosulfate, 0.33 mM pyruvate, 1.7 mM CaCl_2_·2H_2_O, 0.3% (*w*/*v*) fatty acid-free bovine albumin, 1% (*v*/*v*) non-essential amino acid, and 2% (*v*/*v*) essential amino acid. The activated oocytes were subsequently transferred into new mSOF droplets covered with mineral oil and continuously cultivated under the conditions mentioned above. The culture medium containing VC (0, 25, 50, and 100 µg/mL) was changed every 48 h. The rates of cleavage and blastocyst were evaluated at 2 d and 7 d, respectively.

### 2.4. IVF and Embryo Culture

The matured oocytes obtained above were used for IVF by procedures as described [15]. Briefly, frozen bull semen was thawed at 39.0 °C for 30–40 s. The semen was rinsed three times by centrifugation in pre-warmed Dulbecco’s phosphate-buffered saline (D-PBS). Subsequently, the sperm was rinsed in 5 mL TALP [Tyrode’s medium base, albumin, lactate, and pyruvate, containing 0.67 mg NaCl/mL, 0.024 mg KCl/mL, 0.005 mg NaH_2_PO_4_/mL, 0.029 mg CaCl_2_·2H_2_O/mL, 0.01 mg MgCl_2_·6H_2_O/mL, 0.09 mg glucose/mL, 0.01 mg sodium pyruvate/mL, 0.19 μL sodium lactate (60%, *v*/*v*)/mL, 0.21 mg NaHCO_3_/mL, and 0.3 mg BSA/mL] and incubated at 38.5 °C, 5% CO_2_, and saturated humidity for 30 min. During sperm incubation, the COCs cultured for 23–24 h were washed in TALP 3 times, transferred into 90 µL TALP droplets with 25–30 COCs per droplet, and continuously incubated at 38.5 °C, 5% CO_2_, and saturated humidity.

Next, the sperm was washed with TALP 2 × 5 min by centrifugation at 1500 r/min and diluted with TALP to reach a final density of 2–4 × 10^6^/mL. Subsequently, 10 µL sperm was added into 90 µL TALP droplets containing COCs and the sperm-COCs were coincubated for 24 h.

After sperm-COCs coincubation, the putative zygotes were transferred into a pre-warmed operating solution (500 µL). The remaining cumulus cells around the oocytes were removed by pipetting repeatedly. The zygotes were rinsed with mSOF 3 × 5 min and transferred into new mSOF containing VC (0, 20, 40, and 60 µg/mL based on preliminary experiments). Generally, 25–30 zygotes were put into 100 µL mSOF with/without VC. The medium was replaced every 48 h. Similarly, the rates of cleavage and blastocyst in each group were evaluated at cultivation of 2 d and 7 d, respectively.

### 2.5. Blastocyst Cell Count

Four to five blastocysts from the IVF embryos cultured for 7 d were selected for nuclear staining. Briefly, the embryos were rinsed three times with phosphate-buffered saline (PBS)/PVA (0.01% polyvinyl alcohol in PBS). The zona pellucida of the embryos was removed with the acidic Tyrode’s solution [0.8 mg NaCl/mL, 0.02 mg KCl/mL, 0.024 mg CaCl_2_·2H_2_O/mL, 0.0047 mg MgCl_2_·6H_2_O/mL, 0.1 mg glucose/mL, and 0.4 mg polyvinylpyrrolidone (PVP)/mL]. After washing 3 × 5 min with PBS/PVA, the embryos were stabilized using 3.7% paraformaldehyde at room temperature for 3 min, washed 3 times with PBS/PVA, and dyed with Hoechst33342 (10 µg/mL) for 10 min in darkness. Finally, the embryos were washed 3 times with PBS/PVA. Next, they were mounted between a cover slip and a glass slide. The samples were immediately observed under a fluorescence microscope (Nikon, Tokyo, Japan). The number of blastomeres in each blastocyst was counted; the average cell numbers were quantified and statistically compared among groups.

### 2.6. Quantitative Real-Time Polymerase Chain Reaction (qRT-PCR)

The total RNAs of the blastocysts (approximately 25 embryos for each group) were isolated using the DNA/RNA Micro Kit (Qiagen, Hilden, Germany). The RNA was reverse-transcribed into cDNA using TransScript^®^ II One-Step gDNA Removal and cDNA Synthesis SuperMix AH311-03 Kit (Transgen Biotech, Beijing, China). The reverse transcription was performed in a 20 µL system, using the following program: 95 °C for 30 s, 95 °C for 5 s, and 60 °C for 30 s, for a total 40 cycles, adhering to the protocols provided by the manufacturers. The primer sequences were detailed in Table 1 and *18S rRNA* served as the reference gene to normalize the data. The 2^−∆∆CT^ method was employed to analyze the relative abundance of mRNA transcripts.

### 2.7. Statistical Analysis

The cleavage rates = No. of 2-cell embryos/No. of parthenogenetic activation oocytes or IVF oocytes) × 100%] and blastocyst rates = [(No. of embryos with blastocoel/No. of parthenogenetic activation oocytes or IVF oocytes) × 100%]. Data were expressed by the mean ± standard error (SEM) and analyzed statistically using IBM SPSS Statistics 19.0 software. *p* < 0.05 was considered to be a significant difference (* *p* < 0.05, ** *p* < 0.01, and *** *p* < 0.001). Replication of each experimental procedure was conducted three times to ensure reliability.

## 3. Results

### 3.1. Effects of VC on Oocyte IVM and Parthenogenetic Embryo Development

The morphology of bovine oocytes is shown in Figure 1A–C. Each bovine COC contains an immature oocyte with a minimum of three layers of compact cumulus cells (Figure 1A). After IVM 24 h, the COCs were expanded with larger oocytes and loose cumulus cells (Figure 1B). The mature oocytes have intact zona pellucida with clear polar bodies after removing cumulus cells (Figure 1C). The IVM rates of oocytes among four groups were compared, showing that 50 µg/mL VC greatly improved the IVM rate of bovine oocytes (Figure 1D, * *p* < 0.05). No statistical difference was observed for the cleavage and blastocyst rates of the parthenogenetic embryos (Figure 1E, *p* > 0.05). The percentage data were provided in the Appendix A (similar hereinafter).

### 3.2. Effects of VC on IVF Embryo Development

Since IVF embryos might have different responses to VC, its concentration gradient in IVC medium was adjusted to 0, 20, 40, and 60 µg/mL according to our preliminary experiments. The results showed that VC at 20 µg/mL increased the cleavage rate (Figure 2A, *p* < 0.05). For the blastocyst rate, no statistical difference was observed (Figure 2A, *p* > 0.05). Consistently, qRT-PCR analysis demonstrated that the mRNA expressions of the pluripotency-related genes *Oct4*, *Sox2*, *Cdx2*, and *Nanog* in blastocysts were remarkably enhanced by VC at 20 µg/mL (*p* < 0.05 or 0.001). Interestingly, supplementing VC at higher concentrations (40 and 60 µg/mL) did not show positive effects on the expressions of these genes (*p* > 0.05) except *Nanog* (Figure 2B, *p* < 0.05).

### 3.3. Effects of VC Combination on IVF Embryo Development

Accordingly, IVM solution containing 50 µg/mL VC + IVC medium −/+ 20 µg/mL VC (named as 50/0 group and 50/20 group) was applied as a combined strategy to examine the effects of VC on the development of IVF embryos, with none-supplemented IVM solution + IVC medium (0/0 group) as the control. The results revealed that the embryo cleavage rates were greatly increased both in the 50/0 and 50/20 groups (Figure 3A, *p* < 0.05 or 0.01). However, no significant difference in blastocyst formation and the cell numbers in blastocysts was observed (Figure 3A,B, *p* > 0.05). Further qRT-PCR analysis demonstrated that a VC combination of 50/0 and 50/20 dramatically enhanced the transcriptional expression of *Oct4*, *Cdx2*, and *Nanog*, while only VC 50/20 boosted *Sox2* expression in bovine blastocysts (Figure 3C, *p* < 0.05 or 0.01). 

### 3.4. Effect of VC Combination on mRNA Expressions of Apoptotic Genes

Since apoptosis might occur during the processes of oocyte IVM and embryo development, we next investigated the expression of the apoptosis-related genes *Bcl-2* and *Bax* and the changes in the *Bcl-2*/*Bax* ratio in IVF blastocysts derived by varied VC combinations in IVM solution + IVC medium (VC 0/0, VC 0/20, VC 0/40, VC 0/60, VC 50/0, and VC 50/20, Figure 4). Compared with the VC 0/0 group, VC 0/20 and VC 50/20 significantly elevated the transcription levels of *Bcl-2* (*p* < 0.05), while VC 0/40 and VC 0/60 dramatically increased *Bax* expression (*p* < 0.05). However, *Bax* expression was greatly decreased in the VC 50/0 group (Figure 4A, *p* < 0.05). Furthermore, the ratios of *Bcl-2*/*Bax* were analyzed among these groups, showing slight increases in VC 0/20, VC 50/0, and VC 50/20 groups (all > VC 0/0 = 1, although not significantly) and marked decreases in the VC 0/40 and VC 0/60 groups (Figure 4B, *p* < 0.05).

## 4. Discussion

Techniques of IVM, IVF, IVC, and somatic cell cloned embryos have been successfully applied in research, production, animal breeding, and medical fields. Oocyte IVM mainly includes nuclear maturation and cytoplasmic maturation. Many factors can affect oocyte IVM and subsequent embryonic development, such as female age and ovarian function, the temperature and time of ovarian transport, the method for COC aspiration, season, oocyte quality, culture medium, supplements, etc. Optimizing these factors is pivotal to procuring high-quality oocytes and embryos for animal breeding. Since VC is an essential nutrient with multiple functions, we evaluated its effects at an appropriate concentration as well as its combination on the IVM of bovine oocytes and early embryo quality herein.

Previously, in order to optimize embryo culture conditions, researchers applied varied supplements including VC to improve oocyte IVM, IVF rate, and embryo development. It is reported that VC improved the developmental ability of ovine and swine oocytes and embryonic development [16,17]. However, its effects on bovine oocytes and early embryo development are somewhat controversial. Dalvit et al. demonstrated that VC has no significant influence on the maturation of bovine oocytes [18], while Córdova et al. proved that after adding VC to IVM solution for 12 h in vitro, the subsequent development ability and blastocyst formation rate of bovine oocytes were significantly improved [19]. It is also demonstrated that VC increases the cleavage rate of somatic cloned embryos but has no positive effects on the blastocyst rate and even negatively decreases the blastocyst rate and blastocyst quality [20]. We speculate that, for bovine oocyte IVM and embryo development, VC concentration needs to be adjusted appropriately. Based on preliminary experiments, we found that VC at 50 µg/mL (in gradients of 0, 25, 50, and 100 μg/mL) is significantly beneficial to bovine oocyte IVM, while VC at 20 µg/mL (in gradients of 0, 20, 40, and 60 μg/mL) is evidently helpful to improve the cleavage rate of IVF embryos in bovine. As for the blastocyst rate and cell number, the embryos possibly did not respond to the stimulation of VC 20 µg/mL because they might need different environmental factors at different developmental stages. Consistently, supplementing the VC of 20 µg/mL in IVC medium markedly elevated the transcription levels of the pluripotent genes *Oct4*, *Sox2*, *Cdx2*, and *Nanog*. Interestingly, adding VC at a higher concentration into IVM solution (100 μg/mL) or IVC medium (40 and 60 μg/mL, respectively) showed no positive effects on the development of both parthenogenetic and IVF embryos in bovine specimens. These genes are closely connected with the divergent pluripotent states and pluripotency transitions of embryonic cells in mice, pigs, humans, and other mammals, as reviewed previously [11,12]. These results verify that our speculation is reasonable and practical. VC at a tuned concentration might also alleviate cytotoxicity during oocyte IVM as reported in yak [21].

In order to achieve a better effect, an IVM solution containing 50 μg/mL VC + IVC medium containing 20 μg/mL VC (VC 50/20 combination) was employed. Consistently, both VC 50/0 and VC 50/20 showed a statistically positive role in the cleavage rate and a slightly increasing effect on the blastocyst rate. These combination effects were further confirmed by the up-regulated expressions of *Oct4*, *Sox2*, *Cdx2*, and *Nanog* in the IVF blastocysts. It is well documented that these pluripotency-related genes are also important factors in maintaining the proliferation and differentiation of bovine ESCs and iPSCs (induced pluripotent stem cells) [22,23,24,25]. However, whether VC combination in IVM and IVC can regulate the expressions of these genes in bovine embryos is unclear. Thus, our data indicate that an appropriate combination of VC indeed enhances embryo quality by regulating endogenous pluripotency factors of bovine embryos at an early developmental stage.

In the process of embryo IVC, cell apoptosis might occur in its early development due to changes in the culture environment [13]. It is reported that cell apoptosis mainly occurs at the stage of 9–16 cells and compacted morula [14]. It is well known that cell apoptosis is detrimental to embryo survival and developmental competency and eventually leads to decreased rates of cleavage and blastocysts and lower quality of embryos. Therefore, we next asked whether the effects of VC combinations are related to their anti-apoptosis role in bovine embryos. Our data indicate that VC 0/20 and VC 50/20 indeed boost anti-apoptotic gene *Bcl-2* but inhibit pro-apoptotic gene *Bax*. The factors leading to embryo damage are variable, sometimes even including electromagnetic waves. Recently, it was demonstrated that an in ovo VC injection reduces the detrimental influences of electromagnetic waves on chicken embryos [26]. Commonly, exposure to radiation and other detrimental factors raises the possibility of generating more free radicals in embryos and thereby leads to oxidative stress damage. Our results and the above report suggest that VC and its combination prevent embryo damage and death possibly by anti-apoptosis and anti-oxidation activities.

## 5. Conclusions

Obtaining high-quality oocytes and embryos is pivotal for animal breeding, particularly for large livestock like cattle. In this study, the effects of VC on bovine oocyte IVM and early embryo development were evaluated. We found that supplementation of VC at 50 μg/mL into IVM solution is able to improve bovine oocyte IVM and that the combination application of VC 50/20 can potentially change the quality of bovine embryos by boosting pluripotency genes and regulating apoptosis-related genes. Limited by the relatively small sample population in our study, additional experiments may be needed to reveal the underlying mechanism in the next step. From a future view, our results may have practical implications in the IVM and IVF of bovine oocytes as well as the subsequent embryo IVC.

## Figures and Tables

**Figure 1 vetsci-11-00372-f001:**
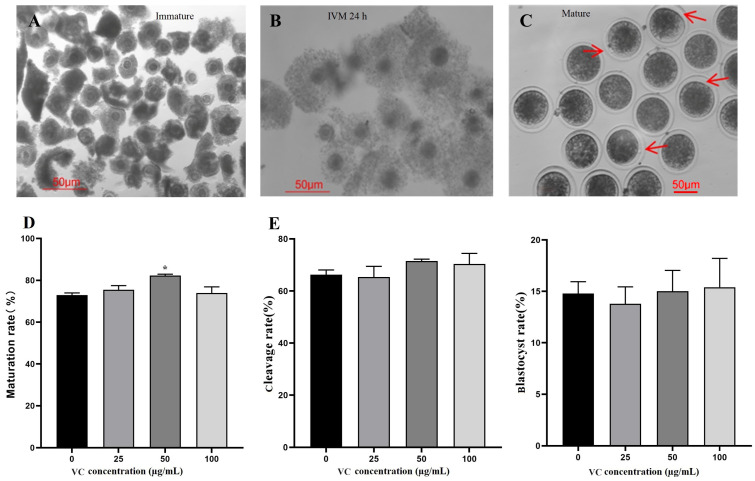
Effects of VC on bovine oocyte IVM and parthenogenetic embryo development. (**A**–**C**): Oocyte morphology at different stages (immature, IVM 24 h, and mature, respectively) was shown with arrows showing the first polar bodies; (**D**): IVM rates (oocyte number = 89, 78, 94, and 81 for different groups, respectively); (**E**): cleavage and blastocyst rates of the parthenogenetic embryos (oocyte number = 102, 96, 111, and 85, respectively). * *p* < 0.05; Scale bar = 50 μm.

**Figure 2 vetsci-11-00372-f002:**
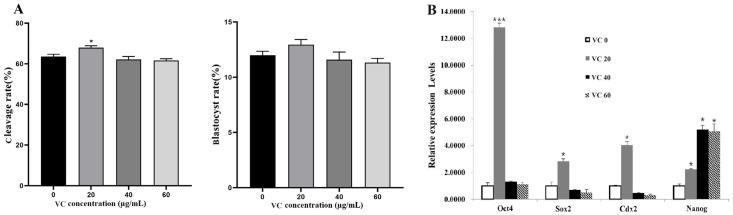
Effects of VC on the development of bovine IVF embryos and the expressions of pluripotency genes. (**A**): The cleavage and blastocyst rates of IVF embryos (oocyte number = 96, 106, 93, and 100 for the different groups, respectively); (**B**): Relative expressions of the pluripotency-related genes *Oct4*, *Sox2*, *Cdx2*, and *Nanog* in IVF embryos. * *p* < 0.05, *** *p* < 0.001.

**Figure 3 vetsci-11-00372-f003:**
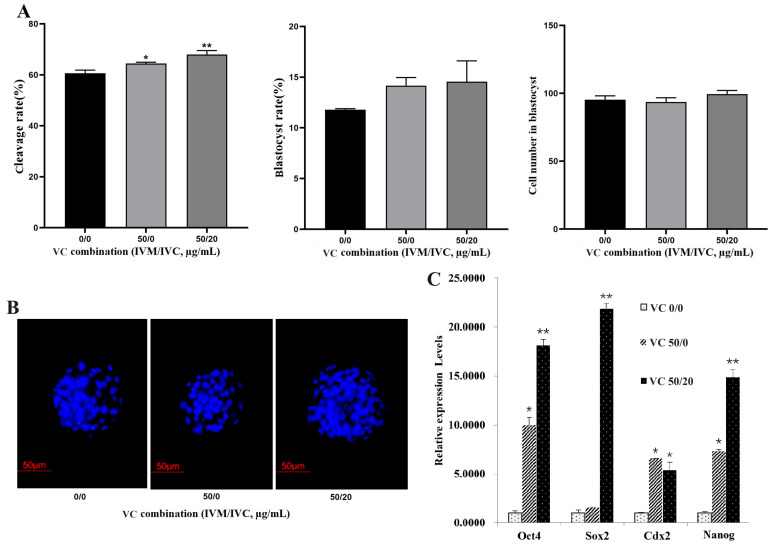
Effects of VC combination on the development of bovine IVF embryo and the expressions of pluripotency genes. (**A**): The cleavage and blastocyst rates as well as the blastomere numbers in IVF embryos (oocyte number = 79, 88, and 95 for different groups, respectively); (**B**): Representative Hoechst33342 staining images of the IVF blastocysts (200×); (**C**): mRNA expression of *Oct4*, *Sox2*, *Cdx2*, and *Nanog* in IVF embryos. * *p* < 0.05, ** *p* < 0.01, Scale bar = 50 μm.

**Figure 4 vetsci-11-00372-f004:**
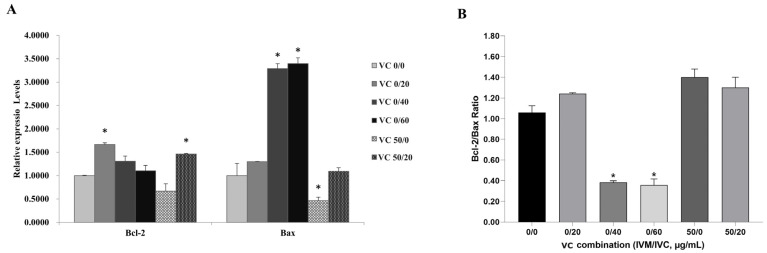
Effect of VC combinations on the mRNA expressions of apoptosis-related genes in bovine IVF blastocysts. (**A**): mRNA levels of *Bcl-2* and *Bax* in groups containing different VC combinations in IVM solution + IVC medium (VC 0/0, VC 0/20, VC 0/40, VC 0/60, VC 50/0, and VC 50/20). (**B**): mRNA expression ratios of *Bcl-2*/*Bax*. * *p* < 0.05.

**Table 1 vetsci-11-00372-t001:** Primers used for qRT-PCR.

Gene	Forward Primer (5′-3′)	Reverse Primer (5′-3′)
*18S rRNA*	GACTCATTGGCCCTGTAATTGGAATGAGTC	GCTGCTGGCACCAGACTTG
*Oct4*	GATTTGGATGAGTTTTTAAGGGTT	ACTCCAACTTCTCCTTATCCAACTT
*Sox2*	CTATGACCAGCTCGCAGA	GGAAGAAGAGGTAACCACG
*Cdx2*	CTTTCCTCCGGATGGTGATA	AGCCAAGTGAAAACCAGGAC
*Nanog*	AAACAACTGGCCGAAGGAATA	AGGAGTGGTTGVTCCAAGAC
*Bcl-2*	GAGTCGGATCGCAACTTGGA	CTCTCGGCTGCTGCATTGT
*Bax*	GCGCATCGGAGATGAATTG	CCACAGCTGCGATCATCCT

## Data Availability

The data that support the findings of this study are available from the corresponding author, upon reasonable request.

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
