# Peer review of "Vitamin C Improves Oocyte In Vitro Maturation and Potentially Changes Embryo Quality in Cattle"

_vetsci, 2024, doi:10.3390/vetsci11080372_

Round 1

Reviewer 1 Report

Comments and Suggestions for Authors

This study has examined the effects of Vitamin C on in vitro bovine oocyte maturation and embryo development.  The authors have identified an optimised level of Vitamin C in the maturation medium and culture medium that has a minor effect on cleavage rates of the embryos.  Expression of pluripotency genes and apoptosis related genes was altered in blastocysts, with differing effects of the various combinations of exposure to Vitamin C.

The overall conclusion indicates that vitamin C in combination in the oocyte maturation medium and culture medium can improve development rates of bovine embryos.  The evidence for this is relatively limited with an increase in day 2 cleavage rate, but no effect on blastocyst development rates.  The conclusions should be revised to more accurately reflect the results of the study.

Results:  Please specify how the development rates have been calculated.  Are the blastocyst rates calculated relative to oocytes matured, or cleaved embryos.    Blastocyst rates are very low.  Is this the usual rate observed in the laboratory?

Figure titles could be reworded to make it clearer for the reviewer that Figure 1 represents Vc in the IVM medium only, while Figure 2 represents Vc in the embryo culture medium only.

Has GAPDH been validated as an appropriate reference gene?  Was it confirmed that GAPDH expression does not change under the conditions used?

Table 1: GAPDH Forward and Reverse primers appear to be the same.

Figure 4:  It is not clear why the groups are joined by a line graph, these were independent groups?

Why such major changes in pluripotency genes are observed at only one concentration of Vc is not clear, and minimally discussed.  Similarly there is little discussion of the significant increases in Bax at higher Vc concentrations.

What is meant by "at tuned" throughout the manuscript.  Is this optimised?

Minor comments:

Line 21 "oocytes of high quality"

Line 23-24:  "but did not affect development of parthenogenetic".  Please change wording

Line 27: Combining.  Please correct spelling.

Line 29:  Point 4) does not seem to be describing combined treatments.  The first group mentioned is Vc 0/20?

Line 38:  "of oocytes"  Please correct.

Line 42: "found to be involved"  Please correct.

Line 54: " Recently, it has been demonstrated"  Please correct.

Lined 58 - has the study on porcine embryos identified any specific functions that are acting to improve developmental competence. "Based on its versatile function" is a very general description.  The introduction would benefit from some more specific details about what has been found with Vc treatment of oocytes or embryos from other species.

Line 78: aspirated - correct spelling.

Line 85: does pieces mean COCs?

Line 88: "after the oocytes have been cultured" Please correct.

Line 166: this paragraph would benefit from restating the criteria used for IVM rate as based on visualisation of a polar body.

Line 230" as well as its"

Comments on the Quality of English Language

The manuscript is generally well written, but requires some editing within the Abstract and Introduction for grammar.

Author Response

Please check the attached Cover letter for revision in which your comments/suggestions/concerns were replied point by point.

Reviewer 2 Report

Comments and Suggestions for Authors

The study shows that vitamin C significantly improves the in vitro maturation rates of bovine oocytes and enhances the expression of pluripotency-related genes in blastocysts at specific concentrations, though its effects on cleavage and blastocyst rates vary. Additionally, certain VC combinations influence the expression of apoptosis-related genes, indicating potential regulatory roles in embryo development.

Major concerns-

The section on the expression of apoptosis-related genes (Bcl-2 and Bax) and the Bcl-2/Bax ratio is somewhat confusing. The results mention significant changes but do not provide detailed quantitative data or statistical analysis for each group. Moreover, the relevance of these findings to the overall study objectives is not clearly articulated. A more thorough explanation of how these changes impact embryo development and viability would be beneficial.

The results for different VC concentrations on cleavage and blastocyst rates are presented without a clear, standardized format. For example, the oocyte numbers are given in the text but not clearly linked to specific groups in the figures. This inconsistency can confuse the reader. Including a summary table with all relevant metrics (e.g., cleavage rates, blastocyst rates, gene expression levels) for all experimental groups would enhance clarity.

Although the results mention statistical significance, the presentation of p-values is inconsistent (e.g., p < 0.05 vs. *P < 0.05)

the reference to "Figure 1 (A-C)" does not provide sufficient detail on what each panel specifically represents.

The text is somewhat repetitive, particularly in the descriptions of the results of cleavage and blastocyst rates across different sections.

While the results indicate changes in gene expression and developmental outcomes, they do not delve into the underlying mechanisms.

Some results are vaguely presented, such as "no statistical difference" or "dramatically enhanced.

Comments on the Quality of English Language

CAN BE IMPROVED

Author Response

(The authors gave the same response as above.)

Reviewer 3 Report

Comments and Suggestions for Authors

This study investigated the effects of vitamin C (Vc) on in vitro maturation of bovine oocytes and early embryo development. According to the authors, addition of 50 μg/mL Vc to IVM medium only improved the oocyte maturation rate, while supplementation of 20 μg/mL Vc during in vitro culture or combing supplementation of 50 μg/mL in IVM medium + 20 μg/mL Vc in IVC medium increased the cleavage rate of IVF embryos and enhanced the expression of pluripotency genes. Overall, this is a well-prepared manuscript. However, I think that there are problems with experimental design and data interpretation. In addition, some of the findings have already been reported in previous studies. Therefore, I strongly suggest that further data should be added as below. If so, this paper would become much more convincing.

Specific Comments:

1) The percentage data (maturate rate, cleavage and blastocyst rate, etc.) should be given in the result section.

2) The differences of maturation, cleavage, and blastocyst rate were not statistically significant, which may be related with the small sample size and the less experimental repetition times. Authors should increase sample size.

3) Result 3.3 should both set 0/20 group.

4) Figure 3, the quality of the representative photos is too poor, which makes the image look awful.

5) Vc is well known for its antioxidant property. The additional experiment would be required for further publication. Examination of ROS and GSH production, or antioxidant-related genes expression in oocyte or embryo would help to elucidate the protective effects of Vc.

6) The discussion does not explain well why Vc supplementation only improved cleavage rate, and the authors should make more hypotheses

Comments on the Quality of English Language

Author Response

(The authors gave the same response as above.)

Reviewer 4 Report

Comments and Suggestions for Authors

In the work, entitled "Vitamin C improves oocyte in vitro maturation and embryo development in cattle" submitted for review, the authors indicate that the addition of ascorbic acid into the culture medium improves fertilization parameters and further development of embryos after in vitro fertilization. Although the experimental part was done very well, the Introduction and Discussion are poorly written. One gets the impression that something was added to the solid core of the work to resemble both of these chapters.

Introduction:

- the role of ascorbic acid has been described very generally,

- the authors did not explain why they chose these and no other genes for research and why they are important for the processes of fertilization and the initial development of blastocysts in vitro,

- no explanation to what extent the genes selected for research refer strictly to cattle and whether they have ever been previously tested in other animals,

- are there any known relationships between ascorbic acid and the expression of the studied genes during the maturation and fertilization processes of oocytes in cattle and other animals.       

The same applies to Discussion chapter. It is superficial and not very convincing, especially if the authors use works on stem cells and chicken embryos as references.

The chapters "Material and Methods" and "Results" show an extremely opposite scientific value. They are carefully prepared and the drawings and signatures are legible. However, I would like to ask you to remove the "ns" superscripts above bars and keep only the asterisk. It would be worth replacing the chemical name - ascorbic acid in the title and text of "Vitamin C" with it and specifying in M&Ms whether the L- enantiomer or a mixture of L-, D- enantiomers was used (found in some synthetic ascorbic acid preparations). I admit I didn't find this in M&Ms. The abbreviation "Vc" is also not particularly successful. Usually the letter V is reserved for V = velocity (speed) and the lower case v = volume. Hence, it is difficult to get used to this text abbreviation.

Major revision

Author Response

(The authors gave the same response as above.)

Round 2

Reviewer 1 Report

Comments and Suggestions for Authors

The authors have addressed many of the comments, but the manuscript continues to suggest that VC improves embryo development, based on limited data.  A small increase in cleavage rate was observed, but no difference in blastocyst rate.  This may relate to the relatively low number of oocytes/embryos studied, as there is a trend.  However, the title of the article, the simple summary and the abstract still conclude that embryo development rates are improved, although no significant differences in blastocyst development rates were observed.  The main findings were increases in oocyte maturation rate, a small increase in cleavage rate, and changes in gene expression for pluripotency and apoptosis related genes.  While these may indicate changes in embryo quality there is no data to specifically demonstrate this.  Similarly, there is no data that specifically demonstrates that apoptosis is inhibited, as now concluded in the abstract.  No difference in cell numbers were observed and number of apoptotic cells in the blastocysts has not been analysed.  Therefore the conclusions have not been fully revised to accurately reflect the findings of the study.

Comments on the Quality of English Language

Some further editing is required.

Author Response

Comments 1: The authors have addressed many of the comments, but the manuscript continues to suggest that VC improves embryo development, based on limited data.  A small increase in cleavage rate was observed, but no difference in blastocyst rate.  This may relate to the relatively low number of oocytes/embryos studied, as there is a trend.  However, the title of the article, the simple summary and the abstract still conclude that embryo development rates are improved, although no significant differences in blastocyst development rates were observed.  The main findings were increases in oocyte maturation rate, a small increase in cleavage rate, and changes in gene expression for pluripotency and apoptosis related genes.  While these may indicate changes in embryo quality there is no data to specifically demonstrate this.  Similarly, there is no data that specifically demonstrates that apoptosis is inhibited, as now concluded in the abstract.  No difference in cell numbers were observed and number of apoptotic cells in the blastocysts has not been analysed.  Therefore the conclusions have not been fully revised to accurately reflect the findings of the study.

Response 1: It's a pity that we didn't get your point in our first round revision. We think about your suggestion very carefully and agree that our current title, simple summary, abstract and conclusions are not very accurate based on our limited data. 

The title was revised as: Vitamin C Improves Oocyte In Vitro Maturation and Potentially Changes Embryo Quality in Cattle

In Simple Summary: the last sentence was revised as "We found that vitamin C supplementation at tuned concentration improved bovine oocyte maturation, and its combination in the media of in vitro maturation and in vitro culture can potentially change the quality of bovine embryos.

In Abstract, the last sentence was revised as: Together, employing 50 µg/mL VC improves the IVM of bovine oocytes, and combination of VC 50/20 potentially changes bovine embryo quality by enhancing the expressions of the pluripotency genes and regulating the expressions of apoptosis-related genes.

In Introduction, minor improvements were also made.

For figures, we deleted all the oocyte number superscripts above the bars according to another reviewer's comments (he/she said it's too overcomplicated). These numbers can be found in figure legends and the supplement tables.

Similar to the revisions for the new title, simple summary and abstract, "5. Conclusions" was greatly revised. Particularly, limitations of our research were point out.

Comments 2: Some further editing is required.

Response 2: We checked the grammar and syntax throughout the manuscript. Hope you now find our paper is suitable for publication in Vet Sci.

Reviewer 2 Report

Comments and Suggestions for Authors

The authors have satisfactorily addressed all comments. The manuscript is now suitable for publication.

Author Response

The authors have satisfactorily addressed all comments. The manuscript is now suitable for publication.

Reply: Thank you so much for encouragement!!!

Reviewer 4 Report

Comments and Suggestions for Authors

In fact, the authors introduced the suggested corrections to the text. At the same time, the grammar and style of the English language in the article have been significantly improved. The substantive level of the article has also increased. However, the authors overcomplicated the superscripts above the bars in figures. Putting the number of attempts (repetitions) there is a bad idea. Their place is primarily in the chapter "Statistical analyses". They can be optionally added in the chart description. Above the error bar there should only be letter markings or asterisks (the system used by the authors) regarding the statistical significance of a given value and nothing else. Please correct it. Minor revision.

Author Response

Comments 1: In fact, the authors introduced the suggested corrections to the text. At the same time, the grammar and style of the English language in the article have been significantly improved. The substantive level of the article has also increased. However, the authors overcomplicated the superscripts above the bars in figures. Putting the number of attempts (repetitions) there is a bad idea. Their place is primarily in the chapter "Statistical analyses". They can be optionally added in the chart description. Above the error bar there should only be letter markings or asterisks (the system used by the authors) regarding the statistical significance of a given value and nothing else. Please correct it. Minor revision.

Response 1: Thank you so much for the encouragement to our revision efforts. As for the oocyte numbers above the bars in figures, we did this according to the comments from one of the Reviewers in first round. These numbers actually have been added in the figure legends and we also provided them in the supplemental tables. Anyway, these superscripts were all deleted this time.

We also further checked the typros and improved the title, simple summary, abstract, and conclusions according to another reviewer's comments.

Hopefully, this time you may find our paper is suitable for publication in Vet Sci.